# Disulfiram/Copper Induces Antitumor Activity against Both Nasopharyngeal Cancer Cells and Cancer-Associated Fibroblasts through ROS/MAPK and Ferroptosis Pathways

**DOI:** 10.3390/cancers12010138

**Published:** 2020-01-06

**Authors:** Yiqiu Li, Fangfang Chen, Jun Chen, Siocheong Chan, Yi He, Wanli Liu, Ge Zhang

**Affiliations:** 1Department of Microbial and Biochemical Pharmacy, School of Pharmaceutical Sciences, Sun Yat-sen University, No. 132 Waihuandong Road, University Town, Guangzhou 510006, China; liyq25@mail2.sysu.edu.cn (Y.L.); chenff29@mail2.sysu.edu.cn (F.C.); chenj279@mail2.sysu.edu.cn (J.C.); chansioch@mail3.sysu.edu.cn (S.C.); 2State Key Laboratory of Oncology in South China, Collaborative Innovation Center for Cancer Medicine, Guangdong Key Laboratory of Nasopharyngeal Carcinoma Diagnosis and Therapy, Sun Yat-sen University Cancer Center, Guangzhou 510060, China; heyi@sysucc.org.cn

**Keywords:** nasopharyngeal cancer (NPC), (disulfiram/copper) DSF/Cu, apoptosis, ferroptosis, cancer-associated fibroblasts (CAFs)

## Abstract

Disulfiram/copper (DSF/Cu) is a promising antitumor reagent for clinical application due to its excellent anticancer activity and safety. However, the anticancer mechanism of DSF/Cu has not been fully elucidated. Our study showed that DSF/Cu strongly induced cytotoxic effects on both nasopharyngeal carcinoma (NPC) cells and α-smooth muscle actin (α-SMA)-positive fibroblasts. Fluorescence activated cell sorting (FACS) analysis further showed that DSF/Cu induced a higher late apoptosis rate in α-SMA-positive fibroblasts than in tumor cells, and DSF/Cu promoted apoptosis and necrosis by an aldehyde dehydrogenase (ALDH)-independent method. Furthermore, we found that the antitumor activity of DSF/Cu against NPC cells occurred through ROS/MAPK and p53-mediated ferroptosis pathways, and that the ROS scavenger *N*-acetyl-l-cysteine (NAC) could reverse the cellular and lipid ROS levels. In 5-8F xenografts, both TUNEL and immunohistochemical (IHC) analyses indicated that DSF/Cu could induce apoptosis and inactivate cancer-associated fibroblasts (CAFs) by inhibiting the expression of α-SMA. In addition, combined with cisplatin (CDDP), DSF/Cu was well tolerated in vivo and could significantly suppress the growth of NPC tissues. Our study demonstrated that DSF/Cu induced antitumor activity against both tumor cells, as well as CAFs and suggested that the use of DSF/Cu as an adjunctive therapy for NPC is worthy of consideration.

## 1. Introduction

Nasopharyngeal carcinoma (NPC) is a malignant head and neck tumor with a distinct racial and geographical distribution that is highly prevalent in Southeast Asia [1,2,3]. Currently, the main treatment for NPC is using radiation therapy or a combination of radiation and chemotherapy. Side effects are more difficult to tolerate, and hearing toxicity is inevitable [4,5]. So, more effective and less toxic drugs are needed.

Disulfiram (DSF), a clinically used anti-alcoholism drug, can inhibit the enzymatic activity of aldehyde dehydrogenase (ALDH) [6]. Over the past decade, increasing evidence indicates that DSF possesses a great potential for the treatment of human cancers [7,8,9]. DSF was previously reported to inhibit the activity of protease, cause the accumulation of polyubiquitinated proteins as well as the aggregation of cytotoxin protein, and lead to cancer cell death [10]. Recent studies showed that DSF/Cu induced NPL4 (an adapter of p97/VCP segregase) aggregation and induced a complex cellular phenotype, finally leading to cell death [11]. Moreover, DSF/Cu was reported to induce significant cell cycle arrest and apoptosis, cause the disturbance of the ROS balance, and activate the stress-related ROS-JNK pathway as well as downregulate the NF-E2-related factor-2 (Nrf2) and Nuclear factor-*к*B (NF-*к*B) pathways in some tumor cells [12,13,14]. In addition, the anticancer effect of DSF is also related to epigenetics [15]. Importantly, DSF can be combined with antitumor drugs to enhance the antitumor effect [16,17]. These studies have highlighted the importance of DSF in antitumor therapy, such as acute myeloid leukemia, prostate cancer, non-small cell lung cancer, breast cancer and so on [13,14,15,16,17]. However, more research is needed on the role of DSF/Cu in NPC.

Ferroptosis is an iron-dependent formation of nonapoptotic cell death, which involves lethal, iron-catalyzed lipid damage [18]. Lipid peroxidation damage on the cell membrane has been shown to be a necessary condition for ferroptosis [19,20]. The cystine/glutamate antiporter inhibitors erastin and sulfasalazine can inhibit the uptake of cystine, cause glutathione peroxidase 4 (GPX4) inactivation and lead to increased accumulation of lipid peroxidation in tumor cells [21]. Furthermore, p53 was also found to be involved in mediating ferroptosis [22,23,24]. Notably, solute carrier family 7 member 11 (SLC7A11), spermidine/spermine N1-acetyltransferase 1 (SAT1) and lipoxygenases (ALOXs) have also been shown to be mediated by p53 [23,24]. In addition, ferroptotic agents can increase the level of death receptor 5 (DR5) and initiate apoptosis [25]. Recent studies showed that targeting ferroptosis-associated metabolism could improve the efficacy of cancer immunotherapy [26].

In the present study, we investigated the antitumor capabilities of DSF/Cu against NPC, and the signalling mechanisms by which DSF/Cu treatment induces NPC cell death. We demonstrated that DSF/Cu was highly cytotoxic to NPC tissues and elucidated the potential antitumor role of DSF/Cu via ferroptosis.

## 2. Results

### 2.1. DSF/Cu Irreversibly Reduces the Viability of NPC Cells

MTT and LDH assays were performed to investigate the cytotoxic effects of DSF/Cu on four NPC cell lines: 5-8F, CNE2, S18 and 6-10B. The nonmalignant nasopharyngeal epithelial cells NP69-SV40T were used as non-tumor cell control. No obvious cell death was observed from 1 μM Cu-treated NPC and control cells (Appendix A), whereas DSF/Cu exhibited a highly cytotoxic activity against these cells in a DSF concentration-dependent manner (Figure 1A, Appendix A). In the four NPC cells lines (5-8F, CNE2, S18 and 6-10B), cell viability was significantly inhibited by treatment with DSF/Cu at 0.2 μM/1 μM, and the inhibition effects reached their peaks at 1.0 μM DSF.

Interestingly, DSF/Cu also showed a vigorous cytotoxicity on the nonmalignant NP69 cells (Figure 1A). What is more, NP69 cells were more sensitive to lower doses of DSF (<0.2 μM) than the NPC cells. Furthermore, MTT and LDH assays (Figure 1B and Appendix A) showed that when treated with a relatively high dose of DSF/Cu (1 μM/1 μM), the reduction of viability was observed in a time-dependent manner, and the inhibition rate was over 80% in these five cell lines at 24 h. These results indicated that DSF/Cu could decrease the cell viability sharply in both tumor and non-tumor cells.

Furthermore, to determine whether the cytotoxic effect of DSF/Cu against NPC cells was reversible, DSF/Cu was removed after 0.5, 1 and 2 h of administration, and then drug-free media were added. As shown in Figure 1C and Appendix A, with 0.5 or 1 h incubation, 5-8F viability decreased significantly after 24 h of DSF/Cu withdrawal. Moreover, with 2 h of DSF/Cu incubation, cell viability after drug withdrawal was similar to those in the non-withdrawal group. Most of the cells died when cell viability was examined at 12 h. These results indicated that the cytotoxicity of DSF/Cu on NPC cells was irreversible.

### 2.2. DSF/Cu Induces Both Apoptosis and Necrosis in NPC Cells by an ALDH-Independent Method

A colony-forming assay was further performed to verify the antiproliferative effect of DSF/Cu in NPC cells. We used 0.2, 0.6 or 1 μM DSF combined with 1 μM Cu to treat 5-8F cells for 10 days. The number of colony-forming cells of the 0.2 μM DSF/Cu group was dramatically decreased compared to the control group (*p* < 0.001). What is more, with a high dose of DSF (>0.6 μM), 5-8F cells almost stopped growing in vitro (Figure 2A).

Next, FACS analysis showed that DSF/Cu (1 μM/1 μM) induced both apoptosis and necrosis in NPC cells in a time-dependent manner. The percentage of apoptotic cells is represented in the upper right and lower right quadrants, and the necrotic cells are represented in the upper left and the upper right quadrant. 5-8F and CNE2 cells that were treated with DSF/Cu underwent apoptosis starting at 2 or 4 h and reached a high apoptosis rate (about 50%) and a high necrosis rate (about 61%) after 10 h post-incubation (Figure 2B). Furthermore, Western blot analysis revealed that DSF/Cu induced the expression of cleaved-PARP1 and cleaved-caspase3 in 5-8F and promoted caspase3 and PARP1 cleavage within 6 h (Figure 2C).

In addition, qRT-PCR and Western blot analysis showed that the expression of ALDH1A1 was absent, whereas the expression of ALDH2 was strong or moderate in all four NPC cell lines (Figure 3A,B). Moreover, ALDH1A1 but not ALDH2 was detected in NP69, and there was no significant change in ALDH1A1 expression after DSF/Cu treatment (Figure 3C). Next, three specific ALDH2 siRNAs were designed to silence the ALDH2 gene expression, and a scrambled siRNA was used as negative control (NC). As shown in Figure 3D,E, all the three siALDH2 sequences were effective in silencing ALDH2 expression in 5-8F and CNE2 cells. Furthermore, there was no obvious difference in the cell viability between DSF/Cu-treated siALDH2-1 (ALDH2^low^) and NC (ALDH2^high^) NPC cells (Figure 3F). Finally, the MTT assay showed that DSF/Cu had a similar killing effect on both ALDH1A1^+^ A549/CDDP (cisplatin) and ALDH2^+^ NPC cells (Figure 3F). These data suggest that DSF/Cu could induce apoptotic and necrotic cell death by an ALDH-independent method.

### 2.3. DSF/Cu Promotes NPC Cell Apoptosis Via ROS/MAPK Pathways

RNA-seq analyses were performed to evaluate the transcriptome changes in 5-8F cells in response to 4 h treatment with DSF/Cu (1 μM/1 μM). More than 58,000 genes were analyzed; 1179 genes were up-regulated while 399 genes were down-regulated (Figure 4A and Appendix A). (Kyoto encyclopedia of genes and genomes) KEGG analysis further showed that transcriptional misregulation in cancer, the p53 signaling pathway, the MAPK signaling pathway, and the ferroptosis pathway were involved in DSF/Cu-induced apoptosis or death (Figure 4B).

The apoptosis-related MAPK pathway was verified first. The mRNA levels of eight MAPK-related genes (CACNA1G, DUSP1, NR4A1, HSPA2, GADD45G, HSPA6, HSPA1B, HSPA1A) were significantly increased after DSF/Cu (1 μM/1 μM) treatment in 5-8F cells (Appendix A). Similar results were also observed in the mRNA expression of four key genes in the MAPK pathway (MAP2K3, MAP2K4, MAP3K1, JUN) (Appendix A). Then, Western blot analysis was used to verify the activation of the JNK and p38 MAPK pathways. As shown in Figure 4C, phosphorylated JNK (p-JNK) and the p-JNK/JNK ratio began to increase after 10 min post-incubation in DSF/Cu-treated 5-8F cells. Similar results were observed in the expression of p-p38 and the p-p38/p38 ratio (Figure 4C). What is more, JNK inhibitor SP600125 and p38 inhibitor SB203580 can block the up-regulation of JNK and p38 in DSF/Cu-treated 5-8F cells (Figure 4D).

Additionally, it has been reported that the expression of ROS is related to the activation of MAPK pathways [27]. As shown in Figure 4E,F, both FACS and High Content screening (HCS) analyses showed that cellular ROS levels increased significantly after being exposed to DSF/Cu (1 μM/1 μM) for 5 h in 5-8F cells, and ROS levels were reversed by addition of a ROS scavenger *N*-Acetyl-l-cysteine (NAC). In addition, HCS analysis showed that the inhibitors SB203580 and SP600125 could also reverse the expression of cellular ROS levels after 5 h post-incubation in DSF/Cu-treated 5-8F cells (Figure 4F and Appendix A). These data demonstrated that DSF/Cu could induce ROS generation and led to NPC cell apoptosis via the activation of ROS/MAPK signaling pathways.

### 2.4. p53-Mediated Ferroptosis Plays an Important Role in DSF/Cu-Induced Cell Death

Next, qRT-PCR analysis was performed to verify whether DSF/Cu induced NPC cell death via the p53 and ferroptosis pathways. As shown in Appendix A, the expressions of seven p53-related genes (p21, Bcl2, GADD45A, GADD45G, FOS, CDKN1A, PMAIP1) and seven ferroptosis-related genes (SAT1, GCLM, DPP4, HMOX1, MAP1LC3B, GLS2, FANCD2) were changed significantly after 5 h post-incubation in DSF/Cu-treated 5-8F cells. Moreover, Western blot analysis showed that both the p53 protein and its downstream targets p21 and BAX were up-regulated, and the p53 inhibitor Pifithrin-α can block the upregulation of p53, p21 and BAX after 6 h post-exposure to DSF/Cu-treated 5-8F cells (Figure 5A).

In addition, the mRNA levels of the ferroptosis marker Ptgs2 and two related genes SAT1 and ALOX15 were detected by qRT-PCR. Ptgs2 was observed to be up-regulated in a time-dependent manner after 1–6 h post-exposure in 5-8F cells (Figure 5B). SAT1 and ALOX15 were also up-regulated after 4 h post-exposure to DSF/Cu in 5-8F cells (Figure 5C,E), and Western blot analysis also showed that SAT1 was up-regulated in 5-8F after 6 h post-exposure to DSF/Cu (Figure 5D). However, no significant change was detected in SLC7A11, ASCL4 or GPX4, which is not associated with the SAT1 pathway [24] (Appendix A).

Notably, lipid peroxidation damage on the cell membrane is an important fatal factor in ferroptosis, so we investigated the effect of DSF/Cu on lipid ROS levels in 5-8F. FACS analysis showed that the lipid ROS level increased in 5-8F after being treated with DSF/Cu for 6 h, and the lipid ROS level was partly reversed by the addition of the ROS scavenger NAC (Figure 5F).

Furthermore, the MTT results showed that DSF/Cu-induced cell death could only be partially reversed by the ROS scavenger NAC, while the JNK inhibitor, p38 inhibitor and p53 inhibitor could not prevent the cytotoxicity (Figure 5G). Taken together, these data indicate that p53-mediated ferroptosis played an important role in DSF/Cu-induced NPC cell death.

### 2.5. DSF/Cu Reduces the Viability of Fibroblasts and Attenuates Fibroblast Activation

Activated fibroblasts characterized by α-smooth muscle actin (α-SMA) expression are considered to be the main cellular constituents of tumor stroma. We further investigated the cytotoxic effects of DSF/Cu on α-SMA-positive human skin fibroblasts (HSF). Similar to NPC and NP69 cells, DSF/Cu exhibited a highly cytotoxic effect on HSF (ALDH1A1^high^) in a DSF concentration-dependent manner and a time-dependent manner by MTT and LDH assays, and the level of ALDH1A1 was not obvious changed after incubation with DSF/Cu for 6 h (Figure 6A,B, Appendix A). DSF/Cu-treated HSF showed cell shrinkage, shortened gradually from being fibrous to spherical, and eventually died (Figure 6C). In addition, both the activated fibroblast marker α-SMA and fibroblast activation protein alpha (FAP-α) were significantly down-regulated when treated with DSF/Cu for 4 h, suggesting that DSF/Cu treatment led to the functional inactivation of HSF (Figure 6D).

Furthermore, FACS analysis showed that DSF/Cu induced apoptosis and necrosis in HSF in a time-dependent manner (Figure 6E). Surprisingly, compared with NPC cells, DSF/Cu induced HSF apoptosis more rapidly and profoundly. The apoptosis rate was about 46% in HSF after treatment with DSF/Cu for 4 h, while a similar apoptosis rate in NPC needed 10 h of treatment (Figure 6F). With 10 h of treatment, the apoptosis rate of HSF reached 73% (Figure 6E,F). Moreover, the cellular ROS levels in HSF increased after treatment with DSF/Cu for 4 h (Figure 6G).

Next, we investigated the inhibitory effects of DSF/Cu on TGF-β1-stimulated mouse fibroblasts NIH 3T3. As shown in Figure 7A and Appendix A, α-SMA protein was hardly detected in untreated 3T3 cells (ALDH2^high^), but increased significantly after being incubated with 20 ng/mL TGF-β1 for 48 h. The toxicity of DSF/Cu was different between 3T3 and TGF-β1-stimulated 3T3 cells. The cell viability of 3T3 cells was 65%, while it was only 43% in TGF-β1-stimulated 3T3 cells after being incubated with DSF/Cu for 24 h (Figure 7B). Similar to HSF, DSF/Cu-treated 3T3 cells showed cell shrinkage, roundness and floating, but TGF-β1-stimulated 3T3 cells exhibited more obvious morphologic changes after being treated with DSF/Cu (Figure 7C). The expression of α-SMA was almost lost after being treated with DSF/Cu for 12 h in TGF-β1-stimulated 3T3 cells (Figure 7D).

These results indicated that activated fibroblasts were more sensitive to DSF/Cu than tumor cells and inactivated fibroblasts. In addition, DSF/Cu could reduce the expression of α-SMA and attenuate TGF-β1-induced fibroblast activation. These results suggest that DSF/Cu is an optimal target for both cancer cells and activated fibroblasts.

### 2.6. Antitumor Activity of DSF/Cu against NPC In Vivo

To explore the anti-NPC activity of DSF/Cu in vivo, the anti-tumor activity was evaluated in a mouse 5-8F xenograft model; the schematic outline for in vivo drug treatment is shown in Figure 8A. Although there was no significant difference between the combination of CDDP and DSF/Cu or DSF/Cu alone in vitro (Appendix A), the tumor volume in the DSF/Cu, CDDP and CDDP/DSF/Cu groups was suppressed by 60.1%, 63.3% and 77.7%, respectively, after two weeks administration, compared to control group (Figure 8B,C). In addition, the use of CDDP inevitably led to adverse effects of weight loss in mice (*p* < 0.01), while there was no significant weight change in the DSF/Cu group compared with the control group (Figure 8D).

In addition, the control group formed normal solid tumors, but the other three groups formed necrotic tumor tissues (Figure 8B). In the DSF/Cu and CDDP groups, only a small number of normally formed tumor cells were found to be present at the tissue margins by hematoxylin and eosin (H&E) assay, and most tumor tissues exhibited a large area of necrotic center (about 80–90%) and obvious nucleus pyknosis (Figure 8E). Furthermore, the histopathological evaluation illustrated that there were more necrotic areas in the tumor tissues of the CDDP/DSF/Cu group (about 95%) than in the CDDP and DSF/Cu alone group, and there were few normal as well as actively dividing tumor cells present, which suggests that CDDP/DSF/Cu synergistically enhanced anti-NPC activity in vivo. IHC for ki67 marker was used to further explore cell proliferation, and the result showed that the number of ki67 positive tumor cells was significantly increased in the control group compared to the other three treatment groups (Figure 8F).

In addition, the apoptosis of fibroblasts was observed in tumor tissues through TUNEL assay. Compared to the control and CDDP groups, it can be obviously seen that there were TUNEL-positive cancer-associated fibroblasts (CAFs) in the DSF/Cu and CDDP/DSF/Cu groups (Figure 9A). The apoptotic indices were significantly increased in the DSF/Cu and CDDP/DSF/Cu groups compared to the control group (57.5% vs. 16%, *p* < 0.001) and the CDDP group (48% vs. 13.3%, *p* < 0.001). What is more, IHC analysis of CAFs by α-SMA showed that CAFs were obviously activated in the control group, and some activated CAFs (α-SMA-positive) could also be seen in the CDDP group. In contrast, the expression of α-SMA in CAFs was negative in the DSF/Cu and CDDP/DSF/Cu groups (Figure 9B). Moreover, double immunofluorescence staining with α-SMA and TUNEL further exhibited that CAFs were inactive and there was apoptosis in the DSF/Cu group and CDDP/DSF/Cu group (Figure 9C). Taken together, these results suggest that DSF/Cu could induce CAF apoptosis and inactive CAFs in vivo.

Moreover, no obvious damage could be observed in the lung, kidney and liver sections with H&E staining among the four groups (Appendix A). Meanwhile, both biochemical and blood routine indices showed that no significant difference was found in the treatment groups compared to the control group (Appendix A). This means that DSF/Cu was well tolerated in vivo. Collectively, these data demonstrate that DSF/Cu was safe and can target both tumor cells and CAFs in vivo. What is more, combing it with CDDP treatment could improve the therapeutic effect in NPC mice.

## 3. Discussion

One of the mechanisms by which DSF plays an antitumor role is that it can combine with metal ions to form complexes [9,10]. In this study, we further confirmed the Cu-dependent cytotoxic effect of DSF on NPC cells. Some research indicated that DSF/Cu achieved the purpose of cancer treatment by inhibiting the growth of ALDH-positive cancer stem cells [28,29]. However, our study showed that DSF/Cu induced NPC cell apoptosis and necrosis by an ALDH-independent method. Recently, Bing Xu et al. indicated that DSF/Cu can activate the ROS-JNK pathway in acute myeloid leukemia (AML) cells [14]. Consistent with this study, we found that DSF/Cu increase cellular ROS levels and activate the apoptosis-related MAPK pathway. However, only a ROS scavenger, but not a JNK or p38 inhibitor, can abolish the antitumor activity induced by DSF/Cu, suggesting that ROS played a dominant role in DSF/Cu-induced apoptosis and necrosis of NPC cells.

Ferroptosis has been indicated as playing a crucial role in suppressing tumor growth [21]. Activation of the MAPK pathway contributes to ferroptotic cancer cell death [29,30]. Our observations, for the first time, demonstrated that DSF/Cu was an inducer for both ferroptosis and apoptosis in NPC cells. Wei Gu et al., reported that p53-mediated transcriptional activation of SAT1 is critical for ROS-induced ferroptosis [24]. What is more, SAT1 can increase the expression of ALOX15, a lipoxygenase that catalyzes the peroxidation of arachidonic acid, which leads to an increase in lipid ROS levels on the cell membrane, eventually resulting in ferroptosis [24]. In accord with these studies, our results showed that DSF/Cu can induce the expression of p53 protein, and the mRNA levels of SAT1 as well as ALOX15 were also increased in NPC cells. Moreover, both cellular ROS levels and lipid ROS levels were increased under the DSF/Cu treatment. These data demonstrate that ferroptosis played an important role in DSF/Cu-induced cell death. Although ferroptosis is distinct from other forms of cell death like apoptosis, we have detected that DSF/Cu can obviously promote NPC cell apoptosis; the use of the ROS scavenger NAC did not completely restore the morphology and survival of NPC cells, suggesting that the synergistic activation of multiple signature pathways contributed to the strong DSF/Cu-induced antitumor activity, and lipid ROS-induced ferroptosis played the major role in DSF/Cu-induced destruction.

It is well known that cancer-associated fibroblasts (CAFs) are crucial to cancer development, progression, and metastasis. CAFs can not only potentially induce chemoresistance acquisition in tumor cells, but can also directly contribute to the suppression of antitumor T-cell responses [31,32,33]. Currently, anticancer therapies are suggested to target both cancer cells and the stromal compartment to improve patient outcomes [34]. In this study, we are excited to find that DSF/Cu can induce apoptosis and inhibit the expression of α-SMA in CAFs. Our NPC xenograft study clearly indicated that, based on the destruction of both CAFs and tumor cells by DSF/Cu, large solid tumors can be turned into necrotic tumor tissues with a α-SMA-negative CAFs. Overexpression of α-SMA-positive CAFs has been reported to predict poor prognosis in NPC [35]. Our study, for the first time, demonstrated that DSF/Cu plays strong antitumor roles by targeting both tumor cells and CAFs. Moreover, the combination therapy of antitumor drugs and DSF/Cu showed an effective improvement in their anti-NPC activity.

Previously, two studies indicated that DSF/Cu induces NPC cell apoptosis by increasing chloride channel-3 protein expression, or by inducing ROS production and decreasing NF-KB-p65 expression [36,37]. These studies, together with our findings, indicate that DSF/Cu induces anti-NPC activity through a joint action of multiple pathways. Surprisingly, we also found that DSF/Cu showed a strong inhibitory effect on nasopharyngeal epithelial cells NP69-SV40T in vitro. The reason for the inconsistent results may be attributed to the lower DSF concentration (0.4–0.8 μM) in above studies. We further explored the effects of DSF/Cu on other immortalized cells, and found that DSF/Cu (1 μM/1 μM) exhibited strong cytotoxic effects on the human colon epithelial cell line NCM460 as well as the human embryonic kidney 293T cell line (Appendix A). We think that although these nonmalignant cell lines have often been used as normal cells in some studies, similar to tumor cells, NP69-SV40T cells essentially acquired an immortalized nature, and the rapidly growing cells are sensitive to ROS. In addition, it has been reported that a significant depression in copper concentration could be found in tumor tissue and in the serum after treatment with anticancer drugs in tumor-bearing mice, while there was no significant change in the liver [38]. However, we checked the copper ion concentration, and no difference was found between tumors and other normal tissues, except for liver tissue (Appendix A). We suspect that this may be due to the addition of additional copper. These results suggest that DSF/Cu could target solid tumor tissues that are rapidly growing and have regenerative capacity.

Taken together, our study revealed that DSF/Cu can strongly promote NPC cell and CAF apoptosis as well as necrosis in vitro and in vivo. ROS-induced ferroptosis played an important role in DSF/Cu-induced cytotoxicity. It is of certain significance to consider DSF/Cu as an adjuvant drug for the treatment of NPC in clinical practice.

## 4. Materials and Methods

### 4.1. Cell Culture and Treatments

The NPC cell lines 5-8F, CNE2, S18 and 6-10B, nonmalignant nasopharyngeal epithelial cells NP69-SV40T, human lung adenocarcinoma multidrug-resistant cells A549/CDDP, human skin fibroblasts (HSF), mouse embryonic fibroblast cell line NIH 3T3, and human nonmalignant colon epithelial cell line NCM460 were cultured in RPMI 1640 medium (Gibco, Carlsbad, CA, USA). Human embryonic kidney 293T cell line was cultured in DMEM medium (Gibco, Carlsbad, CA, USA). All were supplemented with 10% fetal bovine serum (Gibco, Brazil), 100 IU/mL penicillin and 100 μg/mL streptomycin, in a humidified atmosphere of 5% CO_2_ at 37 °C.

Cells were treated with Cisplatin (CDDP), Copper (II) d-gluconate (Cu) and/or disulfiram (DSF) (all from Sigma Aldrich) for the indicated lengths of time. The withdrawal experiment was used to investigate whether the inhibitory effect of DSF/Cu on NPC was reversible. 5-8F cells were plated in a 96-well plate at 7000 cells per well and treated with DSF/Cu for 0.5, 1 or 2 h. The media containing DSF/Cu were then removed, and cells were washed twice with phosphate buffer saline (PBS). New drug-free media were added, and cells were incubated for 12 or 24 h. An MTT assay was used to determine the inhibitory effect.

### 4.2. Cell Viability Assay

Cell viability was measured using the 3-(4,5-dimethylthiazol-2yl)-2,5-diphenyl-tetrazolium bromide (MTT, Sigma-Aldrich, St. Louis, MO, USA) colorimetric dye method and Lactate dehydrogenase release assay (LDH Cytotoxicity Assay Kit, Beyotime Biotechnology, Shanghai, China), respectively. Briefly, NPC, NP69 and HSF cells were plated overnight at 7000 cells per well in 96-well plates. Cells were then treated with DSF/Cu for the indicated lengths of time. The JNK inhibitor SP600125 (SP, Beyotime Biotechnology, Shanghai, China), p38 inhibitor SB203580 (SB, Beyotime Biotechnology, Shanghai China), p53 inhibitor Pifithrin-α (Pif-α, Beyotime Biotechnology, Shanghai, China), and ROS scavenger *N*-acetyl-l-cysteine (NAC, Beyotime Biotechnology, Shanghai, China) were pretreated for 12 h before DSF/Cu was added. For MTT, the viability of cell growth was measured and calculated according to the following formula: cell viability rate (%) = (experimental value A490)/control value A490) × 100%. For LDH, inhibition rate (%) = (control value A490-experimental value A490)/control value A490 × 100%. Three independent experiments were performed, and each was performed in three replicates.

### 4.3. Colony-Forming Assay

The 5-8F cells were plated in 6-well plates at 500 cells/well and treated with the indicated concentrations of DSF as well as 1 μM Cu. Then, the cells were incubated for 10 days. The cells were fixed in 4% paraformaldehyde and stained with crystal violet. The colonies were counted and compared with the control group (1 μM Cu) cells.

### 4.4. Apoptosis Assessment

Cells that were 80% confluent were treated with DSF/Cu (1 μM/1 μM) for the indicated time. Cells were observed under an inverted microscope (Nikon TE 300, Nikon, Tokyo, Japan), harvested in 5 mM EDTA in PBS, washed and resuspended in Annexin-binding buffer, and then stained with Annexin V and propidium iodide (PI), according to the manufacturer’s instructions (Wanleibio, Shenyang, China). Apoptotic cells were analyzed by FACS Calibur (guava easyCyte, Merck KGaA, Darmstadt, German) using a 488 nm excitation and a 519 nm bandpass filter for fluorescein detection, and a 617 nm filter for PI detection. Data were collected from at least 10,000 cells per sample.

### 4.5. Analysis of ROS Production

Cells were treated with DSF/Cu (1 μM/1 μM) for 5 h (cellular ROS level) or 6 h (lipid ROS level) and washed with PBS. Then, cells were incubated with 10 μM DCFH-DA (Beyotime Biotechnology, Shanghai, China) or 5 μM C11-BODIPY 581/591 (Invitrogen, Carlsbad, CA, USA) in serum-free medium at 37 °C for 30 min in a dark place. Cells then were washed with PBS. ROS levels were analyzed on a flow cytometer (guava easyCyte, Merck KGaA, Darmstadt, German) using a 519 nm filter for DCFH-DA fluorescein detection, and a 617 nm filter for C11-BODIPY 581/591 detection. Data were collected from at least 10,000 cells per sample.

The High Content screening (HCS) assay used automate epifluorescence microscopy to assess ROS levels in 5-8F cells that were plated in 96-well plates at 7000 cells/well, and treated with DSF/Cu (1 μM/1 μM) for 5 h. Inhibitors SB203580 (10 μM), SP 600125 (10 μM), and NAC (10 mM) were pretreated for 12 h. Then, the cells were incubated with 10 μM DCFH-DA for 30 min in a dark place. Cells then were washed with PBS. Cellular ROS levels were analyzed on the ArrayScanVTI (Thermo Fisher, Waltham, MA, USA).

### 4.6. siRNA

ALDH2 siRNA and siRNA NC were designed by RiboBio (Guangzhou, China). NPC cells were seeded into 6-well plates at 2.5 × 10^5^ cells/well and incubated overnight to reach 60–70% confluence for transfection. Transfections were performed using Lipofectamine 3000 (Invitrogen, Carlsbad, CA, USA) according to the manufacturer’s protocol. The target sequences used in the siRNA were as follows: siALDH2-1: 5′-GGAGACTTCTTCAGCTACA-3′; siALDH2-2: 5′-GAGCCAACAATTCCACGTA-3′; siALDH2-3: 5′- GCAGCAACCTCAAGAGAGT-3′.

### 4.7. qRT-PCR and RNA Sequencing

Total RNA was extracted using RNAiso Plus (TaKaRa, Dalian, China) according to the manufacturer’s protocol. cDNA was synthesized from total RNA by using PrimeScript™ RT Master Mix (TaKaRa, Dalian, China). qRT-PCR was carried out on the BIO-RAD CFX96^TM^ (BIO-RAD, Shanghai, China) by using TB Green™ Premix Ex Taq™ II (TaKaRa, Dalian, China). The sequences of primers for the qRT-PCR analysis are listed in Appendix A.

RNA sequencing was performed using the Illumina HiSeq X platform. Briefly, about 2 μg of RNA was isolated per sample. Each group had three samples. The quality control was carried out firstly. Then, the cDNA was synthesized, and finally the cDNA was amplified and purified. Hierarchical Indexing for Spliced Alignment of Transcripts 2 (HISAT2) and other methods were used as reference sequence alignment and differential expression analysis. The Gene Ontology database and Kyoto Encyclopedia of Genes and Genomes were used to analyze data. FastQC (version v0.11.8) was adopted to assess the quality of the raw data. Through analysis, significant *P* value, padjust value and fold-change were obtained for differential gene screening, and the screening conditions were a fold change greater than 1.2 times and less than 0.83333 times. The data were deposited in (Gene-Cloud of Biotechnology Information) GCBI.

### 4.8. Western Blot Analysis

Total cell proteins were separated by 10% SDS-PAGE, transferred to polyvinylidene difluoride membranes, and probed with antibodies directed against human PARP, Caspase3, Cleaved-Caspase3, p-JNK, p38, p-p38 and BAX (1:1000, Cell Signaling Technologies, CST, Boston, MA, USA), JNK (1:500, Wanleibio, Shenyang, China), p53 and p21 (1:1000, proteintech, Chicago, IL, USA), ALDH1A1 and ALDH2 (1:100, Boster, Wuhan, China), SAT1 (1:500, Elabscience, Wuhan, China), α-SMA (1:100, Boster, Wuhan, China), and FAP-α (1:400, R&D Systems, Minneapolis, MN, USA). HRP-conjugated anti-rabbit secondary antibodies were used as a secondary antibody (1:5000, EarthOx Life Sciences, Millbrae, CA, USA). Glyceraldehyde-3-phosphate dehydrogenase (GAPDH) protein levels were also determined by using the specific antibody (1:5000, bioworld, St. Louis Park, MN, USA) as a loading control.

### 4.9. In Vivo Tumor Experiments

Four-week-old BALB/c nude male mice were obtained from Nanjing Biomedical Research Institute of Nanjing University, China, and raised under pathogen-free conditions. All animal procedures were authorized by the Sun Yat-Sen University Animal Experimentation Ethics Committee and performed in accordance with the approved guidelines (Ethics number: SYSU-IACUC-2018-000141).

The mice were inoculated subcutaneously with 5 × 10^6^ 5-8F cells. After the tumors grew to 150–200 mm^3^ on average, mice were randomly assigned to four groups (*n* = 5 per group). Drugs treatment began as follows: 1. Control group, i.g. 300 μL PBS per day and i.p. 150 μL PBS per 3 days. 2. DSF/Cu group, i.g. 2 mg/kg Cu each morning and i.g. 150 mg/kg DSF each afternoon. 3. CDDP group, i.p. 5 mg/kg CDDP per 3 days. 4. CDDP/DSF/Cu group, i.g. 2 mg/kg Cu each morning, i.g. 150 mg/kg DSF each afternoon and i.p. 5 mg/kg CDDP per 3 days. Tumor growth and body weight were monitored every 3 days. Tumor growth was measured in three dimensions twice a week by a caliper. Tumor volume was calculated using the following formula: (length × width^2^)/2.

The experiment was terminated after five doses of CDDP, and mice were sacrificed. The tumors were surgically removed and counted. The tissues were fixed in 4% paraformaldehyde solution prepared for H&E staining or TUNEL assays.

### 4.10. Biochemical Indices, Blood Routine Indices and Determination of Copper Ions

The whole blood of mice was obtained by extracting the eyeball blood under anesthesia [39]. The whole blood of mice (containing 15% EDTA-2K) was collected for the measurement of blood routine indices (XN2000, SYSMEX, Kobe, Japan). On the other hand, after the whole blood was collected, the upper serum was collected by centrifugation for the measurement of biochemical indices (c702, Roche, Basel, Switzerland).

The peripheral blood of mice was obtained by extracting the eyeball blood. After the mice were sacrificed, liver, kidney and tumor tissues were obtained by dissection. The copper ion concentration determination experiment was completed by Shiyanjia Lab.

### 4.11. Histology and Immunofluorescence

Paraffin-embedded tissues were dewaxed, rehydrated and rinsed. Antigens were retrieved by heating the tissue sections at 100 °C for 10 min in citrate (10 mmol/L, pH 6.0) solution. The sections were subsequently immersed in a 3% hydrogen peroxide solution for 10 min to block endogenous per-oxidase activity and were incubated with the primary antibody α-SMA (1:200, Boster, Wuhan, China) or the primary antibody ki67 (1:50, Affinity Biosciences, Cincinnati, OH, USA) at 4 °C overnight. The sections were then incubated with an HRP-conjugated anti-mouse secondary antibody (1:1000, EarthOx Life Sciences, Millbrae, CA, USA) or an HRP-conjugated anti-rabbit secondary antibody (1:1000, EarthOx Life Sciences, Millbrae, CA, USA) at room temperature for 2 h. Finally, the signal was developed for visualization with 3, 3′-diaminobenzidine tetrahydrochloride, and all the slides were counter-stained with hematoxylin.

The tissue sections were subjected to H&E staining and TUNEL staining. H&E staining was used for the routine histopathological examination. TUNEL staining of the tissue sections was performed using an In Situ Cell Death Detection Kit (Roche, Indianapolis, IN, USA) according to the manufacturer’s instructions. The slides were counterstained with hematoxylin (blue). Tumor cell nuclei were quantified by randomly counting 10 fields/section. The apoptotic index (percentage of apoptotic nuclei) was calculated as apoptotic nuclei/total nuclei counted × 100.

Double immunofluorescence staining for α-SMA and TUNEL was performed by incubating the tumor sections with 5% albumin from bovine serum (BSA) for 30 min at room temperature followed by the primary antibody α-SMA (1:50, Boster, Wuhan, China) and the use of an In Situ Cell Death Detection Kit (Roche, Indianapolis, IN, USA) according to the manufacturer’s instructions. The slides were sealed with Prolong Gold Antifade Reagent (Invitrogen, Carlsbad, CA, USA, containing DAPI) before fluorescence microscopy analysis.

### 4.12. Statistical Analysis

All analyses were conducted with GraphPad Prism7 (GraphPad Software, San Diego, CA, USA). The results are the means ± standard deviation (SD) of three independent experiments. A *p-*value of less than 0.05 was considered statistically significant.

## Figures and Tables

**Figure 1 cancers-12-00138-f001:**
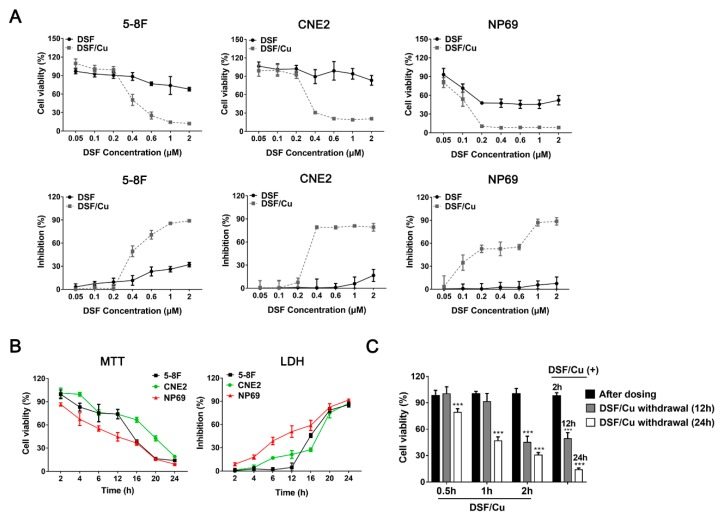
Disulfiram (DSF) reduces the viability of nasopharyngeal carcinoma cells in a Cu-dependent manner. (**A**) The NPC cell lines 5-8F, CNE2 and the nonmalignant nasopharyngeal epithelial cells NP69-SV40T were exposed to the indicated concentration of DSF or with 1 μM Cu for 24 h, after which inhibition effects were determined by MTT assay (above) and LDH assay (below). (**B**) 5-8F, CNE2 and NP69-SV40T were exposed to DSF/Cu (1 μM/1 μM) for different lengths of time, after which inhibition effects were determined by MTT and LDH assays. (**C**) 5-8F cells were exposed to DSF/Cu (1 μM/1 μM) for different lengths of time and were cultured for 12 or 24 h after DSF/Cu withdrawal, after which inhibition effects were determined by MTT assay. DMSO solvent was used as a control. Data are shown as means ± SD. *** *p* < 0.001 vs. after dosing group, *n* = 3.

**Figure 2 cancers-12-00138-f002:**
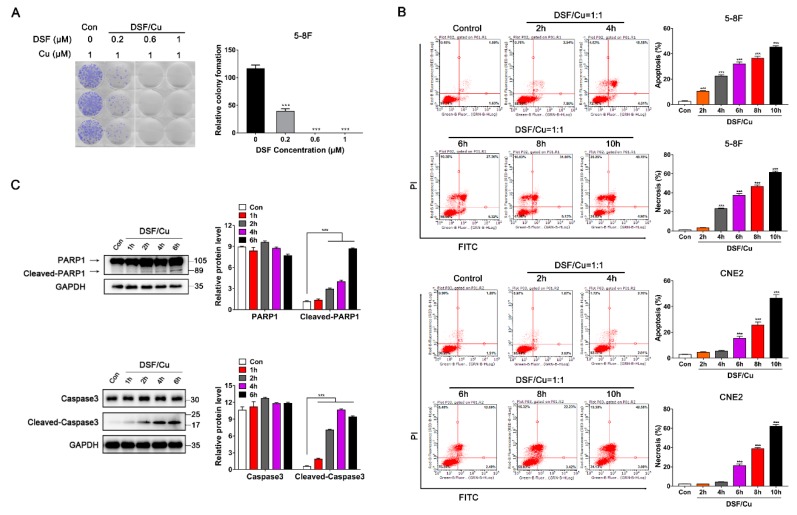
DSF/Cu promotes the apoptosis and necrosis of nasopharyngeal carcinoma cells. (**A**) Representative images and quantification of colony formation assay in 6-well plates. 5-8F cells were incubated for 10 days and the medium containing the drug was replaced once. DMSO solvent containing 1 μM Cu was used as a control. Data are shown as means ± SD. *** *p* < 0.001 vs. control group, *n* = 3. (**B**) Flow cytometry with Annexin V/PI double staining proved that DSF/Cu could significantly increase Annexin V^+^/PI^+^ cells, and promote the apoptosis and necrosis of 5-8F and CNE2. Data are shown as means ± SD. *** *p* < 0.001 vs. control group, *n* = 3. (**C**) Apoptosis-related protein expressions were detected by Western blot in 5-8F, after being cultured with DSF/Cu (1 μM/1 μM) for different lengths of time. Data are shown as means ± SD. *** *p* < 0.001, *n* = 3.

**Figure 3 cancers-12-00138-f003:**
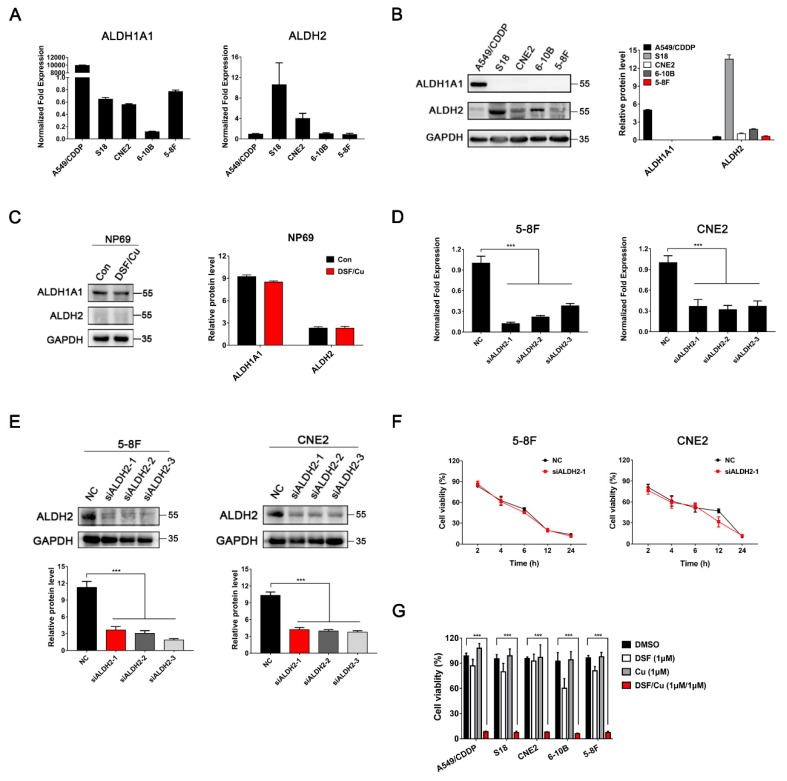
Killing effect of DSF/Cu on tumor cells is ALDH-independent. (**A**) The mRNA expressions of ALDH1A1 and ALDH2 were detected by RT-qPCR; (**B**) ALDH1A1 and ALDH2 protein expressions were detected by Western blot in NPC cell lines and NP69 cells. (**C**) ALDH1A1 and ALDH2 proteins were detected by Western blot in NP69 cells. NP69 cells were exposed to DSF/Cu (1 μM/1 μM) for 6 h. Data are shown as means ± SD, *n* = 3. (**D**,**E**) RT-qPCR and Western blotting were used to measure the expression of ALDH2 in 5-8F and CNE2 cells after transfection with siALDH2 or siNC for 48 h. Data are shown as means ± SD. *** *p* < 0.001, *n* = 3. (**F**) 5-8F and CNE2 cells were exposed to DSF/Cu (1 μM/1 μM) after transfection with siRNA against ALDH2 (siALDH2-1) or control siRNA (NC) for 48 h. After the indicated time points, inhibition effects were determined by MTT assay. (**G**) Cells were exposed to different drugs for 24 h. The inhibition effects were determined by MTT assay. A549/CDDP (cisplatin) was used as ALDH1A1 positive cell control. Data are shown as means ± SD. *** *p* < 0.001, *n* = 3.

**Figure 4 cancers-12-00138-f004:**
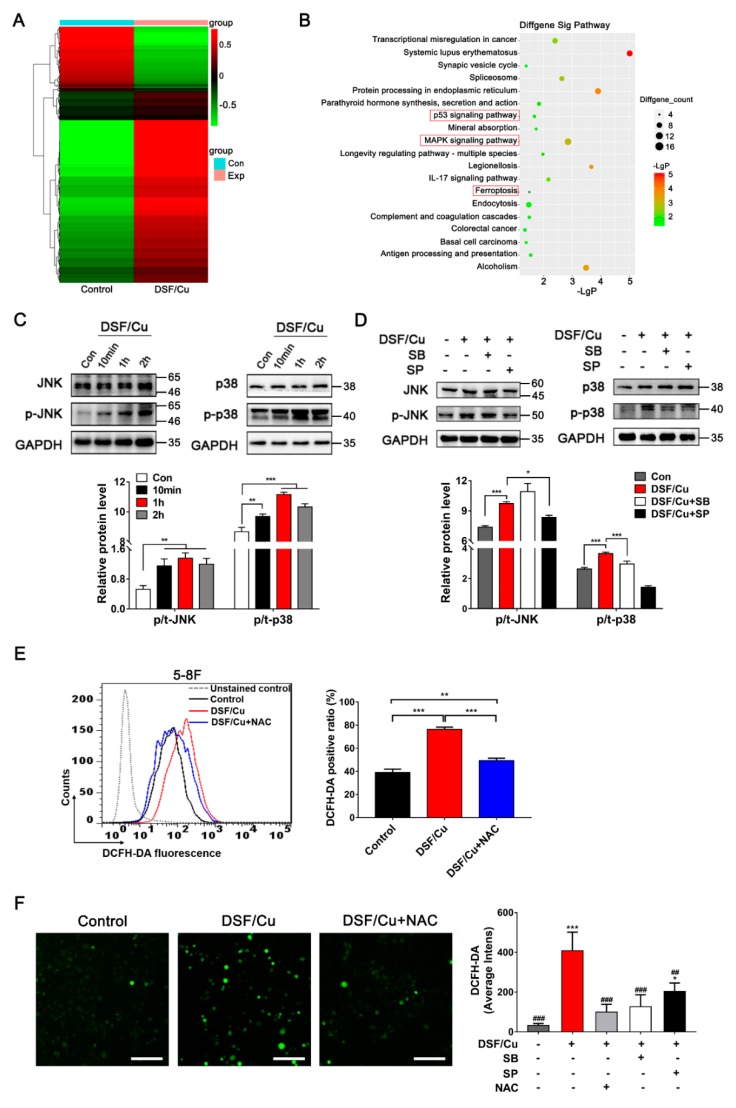
DSF/Cu induces antitumor activity through a joint action of multiple pathways. (**A**) Gene expression heatmap showing differentially expressed genes in 5-8F after being exposed to DSF/Cu (1 μM/1 μM) for 4 h. (**B**) Part of the unregulated significantly differentially expressed gene pathway bubble map; the processing conditions were the same as in (**A**). (**C**,**D**) 5-8F cells were cultured with DSF/Cu (1 μM/1 μM), and the protein expressions were detected by Western blot; the JNK inhibitor SP600125 (10 μM) and p38 inhibitor SB203580 (10 μM) were pretreated 12 h before DSF/Cu was added. Data are shown as means ± SD. * *p* < 0.05, ** *p* < 0.01, *** *p* < 0.001, *n* = 3. (**E**) Cellular ROS production in 5-8F treated with DSF/Cu (1 μM/1 μM) and the ROS inhibitor NAC (10 mM) for 5 h was assessed by flow cytometry using DCFH-DA. NAC was pretreated for 12 h. Data are shown as means ± SD. ** *p* < 0.01, *** *p* < 0.001, *n* = 3. (**F**) Representative images from the High Content screening (HCS) assay of 5-8F cells treated with DSF/Cu, DSF/Cu plus the ROS inhibitor NAC (10 mM) for 5 h. NAC was pretreated for 12 h before DSF/Cu was added. Images with magnification at 10 × are shown here, scale bar: 100 μM. Quantification of cellular ROS levels from the HCS assay are shown as means ± SD. * *p* < 0.05, *** *p* < 0.001 vs. control group. ## *p* < 0.01, ### *p* < 0.001 vs. DSF/Cu group (the second group), *n* = 3.

**Figure 5 cancers-12-00138-f005:**
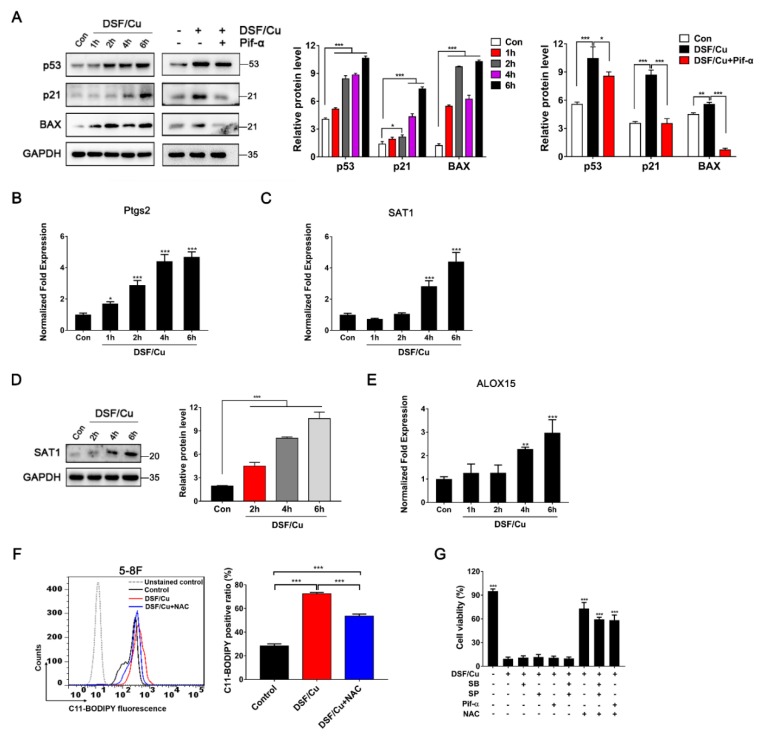
DSF/Cu inhibits nasopharyngeal carcinoma cells via the ferroptosis pathway. (**A**) Western blot analysis for the expression of p53-related proteins in 5-8F treated with DSF/Cu (1 μM/1 μM); the p53 inhibitor Pifithrin-α (20 μM) was pretreated for 12 h. Data are shown as means ± SD. * *p* < 0.05, ** *p* < 0.01, *** *p* < 0.001, *n* = 3. (**B**,**C**) RT-qPCR analysis of Ptgs2 and SAT1 mRNA levels in 5-8F. Data are shown as means ± SD. * *p* < 0.05, *** *p* < 0.001 vs. control group, *n* = 3. (**D**) The expression of SAT1 was detected by Western blotting in 5-8F, after cultured with DSF/Cu (1 μM/1 μM). Data are shown as means ± SD. *** *p* < 0.001, *n* = 3. (**E**) RT-qPCR analysis of ALOX15 mRNA levels in 5-8F. Data are shown as means ± SD. ** *p* < 0.01, *** *p* < 0.001 vs. control group, *n* = 3. (**F**) Lipid ROS production in 5-8F treated with DSF/Cu (1 μM/1 μM) for 6 h was assessed by flow cytometry using C11-BODIPY. NAC was pretreated for 12 h. Data are shown as means ± SD. *** *p* < 0.001, *n* = 3. (**G**) The percentage of viable 5-8F cells after being treated with DSF/Cu (1 μM/1 μM) for 24 h. Inhibitors were pretreated for 12 h. DMSO solvent was used as a control. Data are shown as means ± SD. *** *p* < 0.001 vs. DSF/Cu group (the second group), *n* = 3.

**Figure 6 cancers-12-00138-f006:**
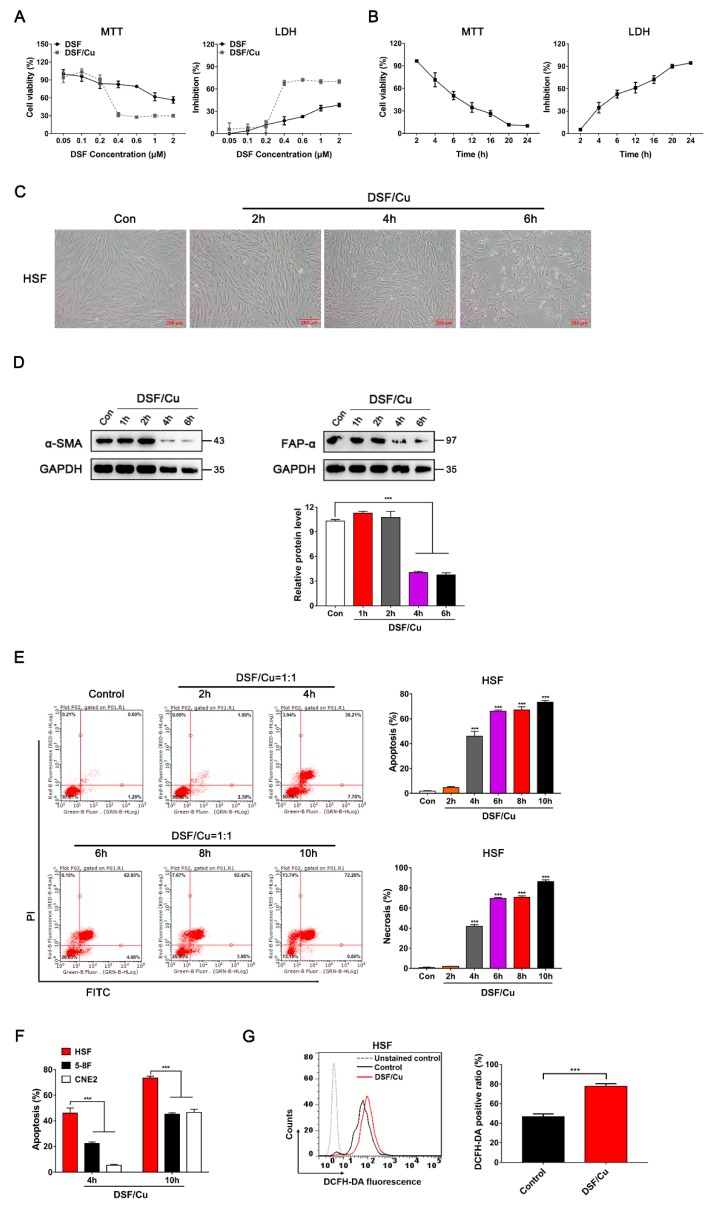
DSF/Cu induces apoptosis and increases the cellular ROS levels in human skin fibroblasts (HSF). (**A**) HSF were exposed to the indicated concentration of DSF or with 1 μM Cu for 24 h, after which inhibition effects were determined by MTT and LDH assays. (**B**) HSF were exposed to DSF/Cu (1 μM/1 μM) for different lengths of time, after which inhibition effects were determined by MTT and LDH assays. (**C**) Representative images of HSF treated with DSF/Cu (1 μM/1 μM) for the indicated time. Scale bar: 200 μM. (**D**) HSF were cultured with DSF/Cu (1 μM/1 μM), and the α-SMA and FAP-α expressions were detected by Western blotting. (**E**) Flow cytometry with Annexin V/PI double staining proved that DSF/Cu could significantly increase Annexin V^+^/PI^+^ cells, promoting the apoptosis and necrosis of HSF. Data are shown as means ± SD. *** *p* < 0.001 vs. control group, *n* = 3. (**F**) The apoptosis rate in HSF, 5-8F and CNE2 after treatment with DSF/Cu for 4 h and 10 h. Data are shown as means ± SD. *** *p* < 0.001, *n* = 3. (**G**) HSF were treated with DSF/Cu (1 μM/1 μM) for 4 h, and cellular ROS production was assessed by flow cytometry using DCFH-DA staining. Quantification of cellular ROS levels is shown as means ± SD. *** *p* < 0.001, *n* = 3.

**Figure 7 cancers-12-00138-f007:**
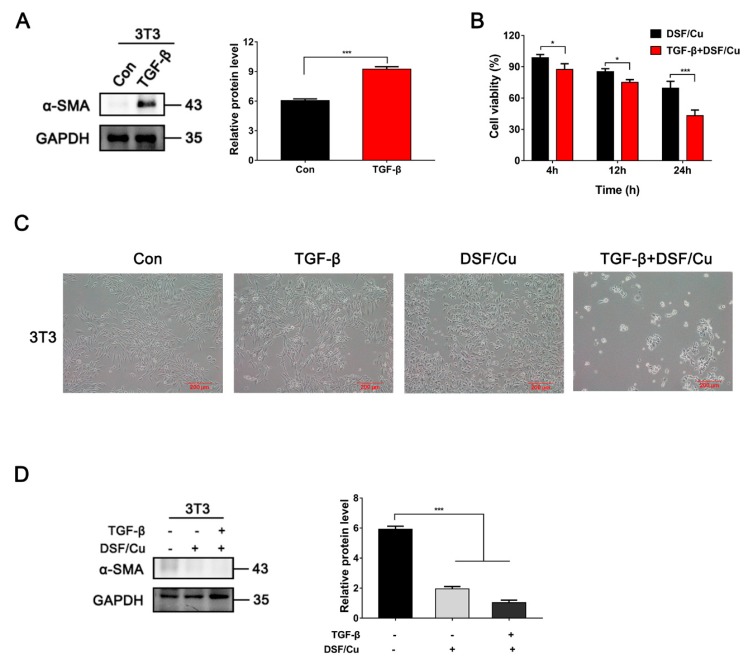
The inhibitory effect of DSF/Cu on TGF-β1 activated 3T3 cells was more obvious than that of normal 3T3 cells. (**A**) α-SMA protein was detected by Western blot in 3T3 cells. TGF-β1 (20 ng/mL) was incubated for 48 h. Data are shown as means ± SD. * *p* < 0.05, *** *p* < 0.001, *n* = 3. (**B**) 3T3 cells were exposed to DSF/Cu (1 μM/1 μM) for different lengths of time, after which, inhibition effects were determined by MTT assay. TGF-β1 (20 ng/mL) was pretreated for 48 h. Data are shown as means ± SD. * *p* < 0.05, *** *p* < 0.001, *n* = 3. (**C**) Representative images of 3T3 treated with DSF/Cu (1 μM/1 μM) for 24 h. TGF-β1 (20ng/mL) was pretreated for 48 h. Scale bar: 200 μM. (**D**) The expression of α-SMA was detected by Western blotting in 3T3 after incubation with DSF/Cu for 12 h. TGF-β1 (20 ng/mL) was pretreated for 48 h. Data are shown as means ± SD. *** *p* < 0.001, *n* = 3.

**Figure 8 cancers-12-00138-f008:**
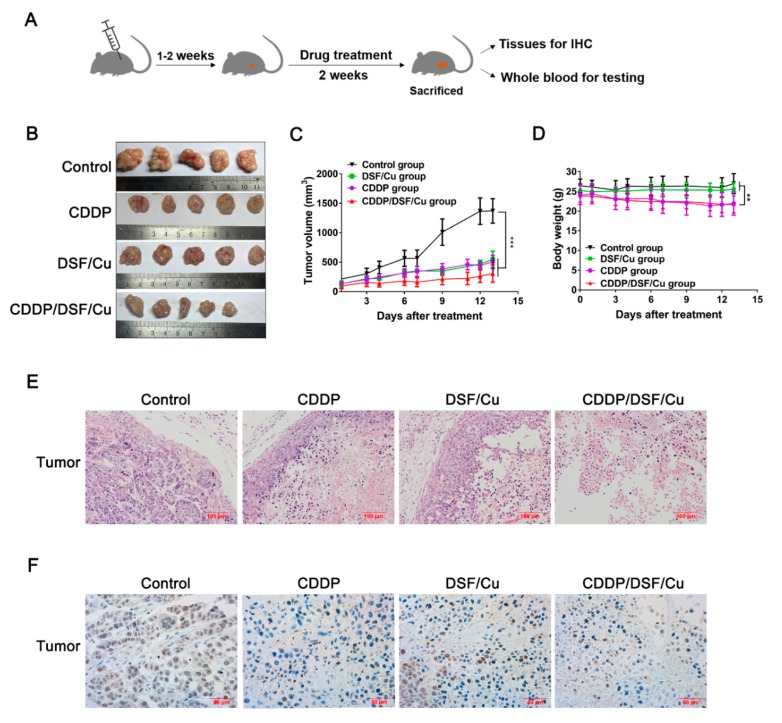
DSF/Cu inhibits the growth of 5-8F xenografts via targeting tumor cells. (**A**) Schematic outline for in vivo drug treatment. (**B**) Photographs of subcutaneously growing human 5-8F xenografts extracted from mice at day 13. (**C**) The growth curves of tumors in mice treated by the indicated drugs. Data are shown as means ± SD. *** *p* < 0.001, *n* = 5. (**D**) Time-course diagram of mice weight. Data are shown as means ± SD. ** *p* < 0.01, *n* = 5. (**E**) Representative hematoxylin and eosin (H&E) staining on tumor biopsies. Scale bar: 100 μM. (**F**) Immunochemical staining of ki67 in tumor tissue slices from the indicated formulation-treated groups. Scale bar: 50 μM.

**Figure 9 cancers-12-00138-f009:**
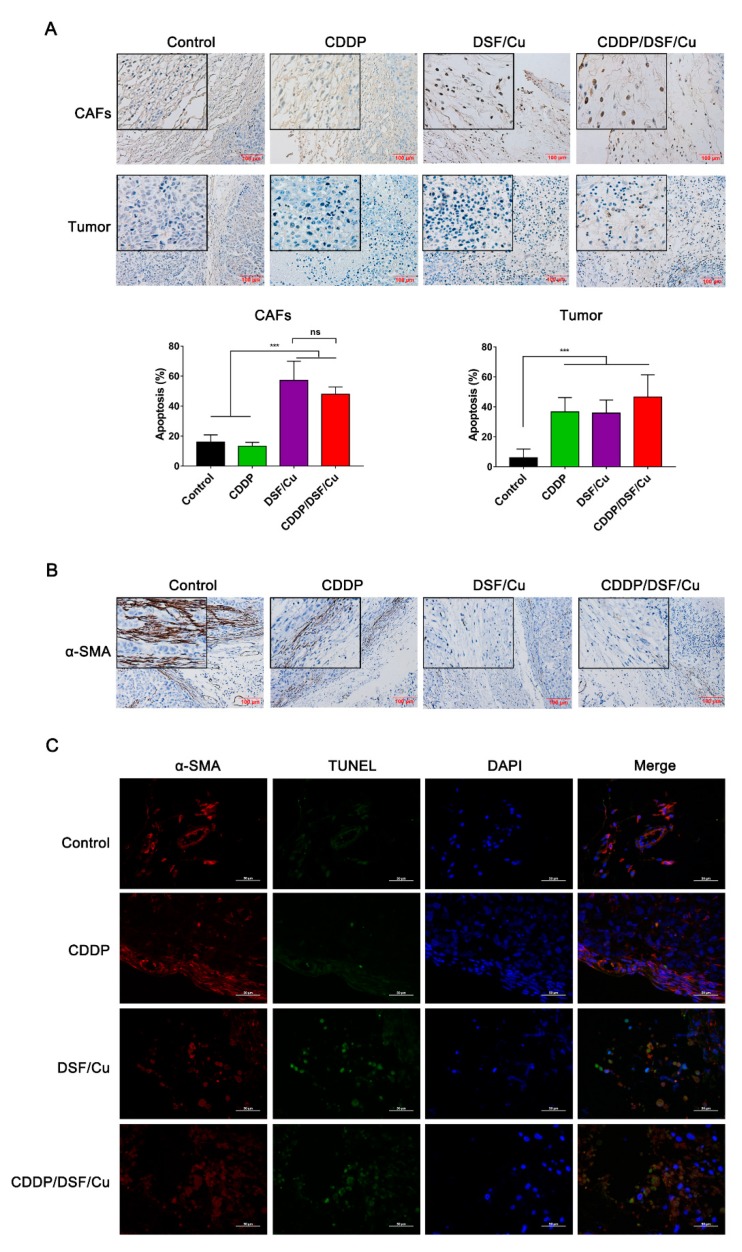
DSF/Cu inhibits the growth of 5-8F xenografts via targeting cancer-associated fibroblasts. (**A**) The TUNEL assay on tumor biopsies. Cancer-associated fibroblasts (CAFs) and the tumor cells in the middle are shown separately. Scale bar: 100 μM. The apoptotic indices (percentages of apoptotic nuclei) are shown as means ± SD. *** *p* < 0.001. (**B**) Immunochemical staining of α-SMA in tumor tissue slices from the indicated formulation-treated groups. Scale bar: 100 μM. (**C**) Double immunofluorescence staining of a-SMA (red color) and TUNEL (green color) and counterstaining of chromatin with DAPI (blue color). Scale bar: 50 μM.

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
