# Peer review of "Disulfiram/Copper Induces Antitumor Activity against Both Nasopharyngeal Cancer Cells and Cancer-Associated Fibroblasts through ROS/MAPK and Ferroptosis Pathways"

_cancers, 2020, doi:10.3390/cancers12010138_

Round 1

Reviewer 1 Report

I am satistified with the modifications introduced in the newest version of the manuscript and the answers to my comments.

I would nonetheless recommend to:

1- correct the M&M regarding the RNA-sequencing analysis. The description of the protocol is not clear (ex: FastQC before cDNA synthesis???).

2- include most of the comments from the "reviewer response letter" into the discussion section of the manuscript. I believe that the discussion of similarities and differences with previous publications must appear in the manuscript.

Reviewer 2 Report

Remarks to the Authors:

In the manuscript entitled “Disulfiram/copper induces antitumor activity against both nasopharyngeal cancer cells and cancer-associated fibroblasts through ROS/MAPK and ferroptosis pathways”, Yiqiu L. et al reported that DSF/Cu induced a cytotoxic effects on both nasopharyngeal carcinoma cells and α-SMA-positive fibroblasts inducing apoptosis and necrosis by an ALDH-independent method. 
Morover, they suggested that the 
antitumor activity of DSF/Cu against NPC cells occurred through ROS/MAPK and p53- mediated ferroptosis pathways.
The manuscript raises an interesting example of adjunctive therapy for NPC. The methods are appropriated and properly conducted. The article is well presented and the quality is adequated. The figures are well presented and the technical quality is good.

I'd ask the authors to take into account the few points raised below:

Could the authors show p-MK2, the direct target of p38 activation (fig.4)? The results showed in fig.4F and 5G should be also supported by genetic ablation of genes to exclude possible off-target effects.

Overall rating

The manuscript is recommended for publication with minor revision.

Author Response

This manuscript is a resubmission of an earlier submission. The following is a list of the peer review reports and author responses from that submission.

Round 1

Reviewer 1 Report

In the paper submitted to cancers MDPI titled: “Disulfiram/copper induces antitumor activity against both nasopharyngeal cancer cells and cancer associated fibroblasts through ROS/MAPK and ferroptosis pathway” by Yiqiu Li et al., authors present a study of a combined disulfiram/Cu effects on cancer cells proliferation and apoptosis induction. In addition, data from RNA seq and measurements of intracellular level of free radicals present an evidence for ROS free radicals accumulation and apoptosis induction together with activation of ferroptosis pathway. Involvement of MAPK and p53 activation is evident. The paper is well organized and in general presents correct data interpretation. However there are several misconceptions or lack of sufficient experimental evidence provided to support the final conclusions. Therefore it is recommended to accept this manuscript, but after major revisions. Here are problems that need to be addressed prior publication in order to improve paper message and its impact.

In Figure 3D authors show that siRNA mediated downregulation of disulfiram target gene- ALDH2 did not sensitize NPC cells to disulfiram/copper combination. This Figure is missing data on normal epithelial cells NP69, since ALDH1/2 expression might still contribute to cell death induced by DSF/Cu in normal cells. This could be an important information for distinguishing between DSF/Cu effects on normal versus cancer cells. In addition, level of ALDH1/2 should be measured in 3T3 mouse fibroblasts after treatment with TGF-b1. Authors provide the evidence for MAPK and p53 activation that is pre-requisite for DSF/Cu induced apoptosis, however pre-treatment with either inhibitor of JNK or p38MAPK did not rescue cells from drug induced apoptosis. Only pretreatment with free radicals scavenger NAC was able to decrease growth inhibition of NPC cells induced by DSF/Cu treatment (Figure 5G). In addition, authors in Materials and Methods talk about pre-treatment with p53 inhibitor Pifithrin a, Pif-a, but in Figure 5G there is no Pif-a, treatment for growth rescue experiment. In Results section however authors suggest that there is dependence on p53 mediated apoptosis. This should be clarified with provided data. P53 dependent gene expression alone is not sufficient evidence for p53 induced apoptosis, growth inhibition assay induced by DSF/Cu with cell lines lacking p53 function could provide necessary information supporting p53 role in DSF/Cu induced apoptosis or ferroptosis. Thus, it is inconclusive if either MAPK and p53 play a significant role in apoptosis or ferroptosis induction in NPC cells. Growth inhibition induced by DSF/Cu in NPC cancer cell line deficient in p53 would provide with additional evidence for p53 involvement in DSF/Cu induced cell death. Authors admit that normal epithelial cell line is very sensitive to DSF/Cu, but they do not provide any evidence for mechanism of action of DSF/Cu on normal cells. The cytotoxicity observed in normal cells, might be related to proliferation status, or status of ALDH1A1, ALDH2 enzymes that could be targeted in normal cells versus cancer, and explain increased toxicity in normal cells versus NPC cells. The level of ALDH1A1 and ALDH2 in NP69-SV40T, HSF or 3T3 cells could be easily measured by WB. In Figure 6 human skin fibroblasts (HSF) are provided as another example for normal fibroblasts sensitivity to DSF/Cu treatment. This Figure is missing ALDH1/2 protein level, since it is a target for DSF. Figure 7D is missing level of aSMA after TGFb1 treatment, a starting point prior treatment with DSF/Cu. In order to conclude that DSF/Cu decreases aSMA protein level it is necessary to include a control with 3T3 cell pre-treated with TGFb1 prior treatment with DSF/Cu. Figure 8 is missing a schematic outline for in vivo drug treatment: starting point and fallowing treatments with end of experiment. In Figure 8B the differences between tumors growth between CDDP and CDDP/DSF/Cu groups of animals are minimal, please comment on drug additive effect on inhibition of tumor growth. In addition, please comment if there is any significant difference between CDDP treatment and treatment with DSF/Cu alone. Authors in the Results section mention TUNEL assay and an assay that measures tumor proliferation. Figure 8 includes growth curve for tumors, but it is indirect measure of proliferation. Figure 8 could provide as a proliferation marker, IHC staining for Ki67. TUNEL is shown in Figure 8E is not specific to fibroblasts, in order to distinguish fibroblasts from tumor tissue, fibroblast specific marker should be used for either double staining or staining of corresponding tissue. The resolution of pictures is not sufficient enough to see differences between CTRL and drug treated samples. Please provide higher magnification pictures that could provide necessary evidence for increase in apoptosis induced by CDDP/DSF/Cu treatment. In Materials and Methods section there is paragraph on both Immunohistochemistry and Histology, both sections appear to overlap in a description of processing the tumor tissues. In addition, in the Histology paragraph, there is a description of quantification of stained positive nuclei for TUNEL assay. Figure 8 does not include any quantification represented in graph. Please correct it appropriately. In the Materials and Methods in paragraph titled: “Biochemical indexes, blood routine indexes and determination of copper ion” authors talk about whole blood draw from removed eye and determination of copper concentration in different tissues. In the supplementary file provided, in Table S2 and S3 there is biochemical data on blood content. Please comment if blood draw from eyeball was done according with authors institutional guidelines and under anesthesia/ analgesia, in accordance with animal welfare standards. Unnecessary suffering of animals and procedures done on animals without adhering to institutional guidelines would preclude the acquired data from publication. In general paper would benefit from careful editing and proofreading. It is recommended that a native speaker would correct this manuscript. Manuscript in part lacks the clarity that might relate to language use.

Reviewer 2 Report

The manuscript by Li and co-authors present several experiments demonstrating the antitumor activity of Disulfiram chelated with Cu2+ (DSF/Cu) against a panel of nasopharyngeal cancer cells (NSC) and against cancer-associated fibroblasts. In addition, authors investigate the mode of action of DSF/Cu in NSC cells. Their findings suggest that DSF/Cu activity is dependent on ROS/MAPK signaling and p53-ferroptosis pathway.

I don’t find the data very supportive of the main conclusions and the overall clarity of the manuscript need to be improve. In addition, two manuscripts by Xu et al. (DOI: 10.1016/j.biopha.2019.109529) and Yang et al. (DOI : 10.1016/j.biocel.2017.01.007) report the antitumor activity of DSF/Cu on NSC. Importantly these latter studies suggest alternative mechanisms to explain DSF/Cu activity and these findings are not explored or discussed in the manuscript.

Major concern 

1) The manuscript does not clarify the mechanism of action of DSF/Cu. It merely describes the correlation between DSF/Cu exposure and the activation of several pathways, including ROS/MAPK signaling and p53-ferroptosis. There is no clear demonstration of a causative relationship between DSF/Cu treatment, ROS/MAPK/ferroptosis activation and cell death. What would happen if these pathways are inactive? In addition, the implication of other pathways described by Xu et al or Yang et al are not investigated or discussed and may be important confounding factors in several experiments.

2) Authors observed that non cancerous NP69 cells are sensitive to DSF/Cu, which is not observed in Xu et al. (DOI: 10.1016/j.biopha.2019.109529). Please discuss.

3) Figure 3 is confusing. Are the cells treated with DSF/Cu in all panels from A to E?

4) In Figure 3 it could be informative to test NP69, human CAF and NIH3T3 cells in the different assays to evaluate the correlation with ALDH-1 and -2 expression levels on a larger pool of cells. In addition, NIH/3T3 cells appear resistant to DSF/Cu (Fig 7B) suggesting that levels of ALDH-1 and -2 must resemble those observed in A549/CDDP cells, if the model is right.

5) It is not explained why experiments are performed at different times after DSF/Cu addition to the cell. For instance, authors conclude that DSF/Cu acts on MAPK signaling, but how? MAPK (JNK and p38) activity is analyzed at 2 hours; RNA-sequencing is performed at 4 hours while RT-PCR validation of « MAPK » genes is performed at 5 hours. It does not appear to be logical and the hierarchy of events following DSF/Cu is thus unclear.

6) The RNA-sequencing analysis unveil a MAPK signature but none of the validated genes are MAPK gene or classical MAPK regulators. In addition, the p-value p=10e-2 and enrichment values are not satisfactory to consider this class of genes for downstream analysis (same is true for ferroptosis and p53 signaling). 

7) In Figure 8, control cells are generating normal solid tumors while the three other groups form cystic tumors. Is it thus not clear how to compare the different groups and thus interpret these data. In addition, the combination of DSF/Cu + CDDP has no clear beneficiary effects compare to DSF/Cu of CDDP alone.

Other points:

- The manuscript need to be properly edited. Too many typos.

- RT-PCR validation are not conclusive. Only 1 gene over 5 in the p53 signature and 1 gene over 4 in the ferroptosis signature are confirmed. In addition, RT-PCR data are discordant for SAT1 between FigS4B and Fig5C.

- In Figure 2B, it would be nice to present a quantification of necrosis, as well as clearly define the quadrants used to score apoptosis and necrosis.

- Most western blot are not convincing, including Fig2C (i.e. cleaved-caspase 3), Fig4C, Fig 4D, Fig5D and Fig7D.

- Fig 4D. What is the concentration of DSF/Cu? Why is JNK not induced as in Fig4C?

- Is SAT1 abundance regulated in a DSF/Cu concentration-dependent manner?

- Provide sequences of siRNAs.

- Please describe the conduct and analysis of the RNA-sequencing analysis. How many samples in each group? How are differentially expressed genes identified (software? P-value? Cut-off?)? Where are the data deposited?

- Figure 4A and 4B. Modify the panels so that information are readable.